# The Burden of Carbapenem-Resistant *Acinetobacter baumannii* in ICU COVID-19 Patients: A Regional Experience

**DOI:** 10.3390/jcm11175208

**Published:** 2022-09-02

**Authors:** Giorgia Montrucchio, Silvia Corcione, Tommaso Lupia, Nour Shbaklo, Carlo Olivieri, Miriam Poggioli, Aline Pagni, Davide Colombo, Agostino Roasio, Stefano Bosso, Fabrizio Racca, Valeria Bonato, Francesco Della Corte, Stefania Guido, Andrea Della Selva, Enrico Ravera, Nicoletta Barzaghi, Martina Cerrano, Pietro Caironi, Giacomo Berta, Cecilia Casalini, Bruno Scapino, Michele Grio, Massimiliano Parlanti Garbero, Gabriella Buono, Federico Finessi, Simona Erbetta, Paola Federica Sciacca, Gilberto Fiore, Alessandro Cerutti, Sergio Livigni, Daniela Silengo, Fulvio Agostini, Maurizio Berardino, Mauro Navarra, Silvia Vendramin, Enzo Castenetto, Marco Maria Liccardi, Emilpaolo Manno, Luca Brazzi, Francesco Giuseppe De Rosa

**Affiliations:** 1Department of Surgical Sciences, University of Turin, 10126 Turin, Italy; 2Department of Anaesthesia, Critical Care and Emergency—Città Della Salute e Della Scienza Hospital, Corso Dogliotti 14, 10126 Turin, Italy; 3Department of Medical Sciences, Infectious Diseases, University of Turin, 10126 Turin, Italy; 4Division of Geographic Medicine, Tufts University School of Medicine, Boston, MA 02111, USA; 5S.C. Anestesia e Rianimazione, Ospedale Sant’Andrea, 13100 Vercelli, Italy; 6S.C. Anestesia e Rianimazione, Ospedale SS. Trinità—Borgomanero—ASL NO, 28021 Borgomanero, Italy; 7S.C. Anestesia e Rianimazione, Ospedale Cardinal Massaia, 14100 Asti, Italy; 8S.C. Anestesia e Rianimazione, Ospedale SS. Arrigo e Biagio, 15121 Alessandria, Italy; 9Department of Translational Medicine, Maggiore della Carità Hospital, University of Eastern Piedmont—UPO, 28100 Novara, Italy; 10S.C. Anestesia e Rianimazione, ASL CN2, 12060 Verduno, Italy; 11Dipartimento di Emergenza ed Aree Critiche, SSD Rianimazione, A.S.O.S. Croce e Carle, 12100 Cuneo, Italy; 12S.C.DU Anestesia e Rianimazione, AOU S. Luigi Gonzaga, Dipartimento di Oncologia, Università degli Studi di Torino, 10043 Orbassano, Italy; 13S.C. Anestesia e Rianimazione, Ospedale di Ivrea, ASL TO4, 10015 Ivrea, Italy; 14S.C. Anestesia e Rianimazione, Ospedale di Rivoli, 10098 Rivoli, Italy; 15S.C. Rianimazione Generale, AO Ordine Mauriziano, 10128 Turin, Italy; 16S.C. Anestesia e Rianimazione Moncalieri-Carmagnola, ASL TO5, 10023 Chieri, Italy; 17S.C Anestesia e Rianimazione Ospedale S. Giovanni Bosco, ASL Città di Torino, 10144 Turin, Italy; 18S.C. Anestesia e Rianimazione, Presidio CTO, AOU Città della Salute e della Scienza, 10126 Turin, Italy; 19S.C. Anestesia e Rianimazione, Ospedale Martini, ASL Città di Torino, 10149 Turin, Italy; 20S.C. Anestesia e Rianimazione, Ospedale di Chivasso, ASL TO4, 10034 Chivasso, Italy; 21S.C. Anestesia e Rianimazione, Ospedale Maria Vittoria, ASL Città di Torino, 10144 Turin, Italy

**Keywords:** *Acinetobacter baumannii*, *Acinetobacter* infections, intensive care unit, COVID-19, SARS-CoV-2, nosocomial infections, carbapenems, multidrug resistance, antimicrobial drug resistance, critical care

## Abstract

Since the beginning of the COVID-19 pandemic, the impact of superinfections in intensive care units (ICUs) has progressively increased, especially carbapenem-resistant *Acinetobacter baumannii* (CR-Ab). This observational, multicenter, retrospective study was designed to investigate the characteristics of COVID-19 ICU patients developing CR-Ab colonization/infection during an ICU stay and evaluate mortality risk factors in a regional ICU network. A total of 913 COVID-19 patients were admitted to the participating ICUs; 19% became positive for CR-Ab, either colonization or infection (*n* = 176). The ICU mortality rate in CR-Ab patients was 64.7%. On average, patients developed colonization or infection within 10 ± 8.4 days from ICU admission. Scores of SAPS II and SOFA were significantly higher in the deceased patients (43.8 ± 13.5, *p* = 0.006 and 9.5 ± 3.6, *p* < 0.001, respectively). The mortality rate was significantly higher in patients with extracorporeal membrane oxygenation (12; 7%, *p* = 0.03), septic shock (61; 35%, *p* < 0.001), and in elders (66 ± 10, *p* < 0.001). Among the 176 patients, 129 (73%) had invasive infection with CR-Ab: 105 (60.7%) Ventilator-Associated Pneumonia (VAP), and 46 (26.6%) Bloodstream Infections (BSIs). In 22 cases (6.5%), VAP was associated with concomitant BSI. Colonization was reported in 165 patients (93.7%). Mortality was significantly higher in patients with VAP (*p* = 0.009). Colonized patients who did not develop invasive infections had a higher survival rate (*p* < 0.001). Being colonized by CR-Ab was associated with a higher risk of developing invasive infections (*p* < 0.001). In a multivariate analysis, risk factors significantly associated with mortality were age (OR = 1.070; 95% CI (1.028–1.115) *p* = 0.001) and CR-Ab colonization (OR = 5.463 IC95% 1.572–18.988, *p* = 0.008). Constant infection-control measures are necessary to stop the spread of *A. baumannii* in the hospital environment, especially at this time of the SARS-CoV-2 pandemic, with active surveillance cultures and the efficient performance of a multidisciplinary team.

## 1. Introduction

The Gram-negative aerobic bacillus *Acinetobacter baumannii (A. baumannii*) primarily causes hospital-acquired infections in especially fragile patients with prolonged hospitalization and with long-term exposition to broad-spectrum antibiotic treatment. It is characterized by disinfection resistance, and its high pathogenicity is increased by the production of a polysaccharide capsule and by the ability to form biofilms [1]. Furthermore, due to the acquisition of multiple antimicrobial resistance, especially to carbapenems, it has been recognized as a major public health concern [2] and considered as “priority 1” (critical) in the World Health Organization (WHO)’s first list of “priority pathogens” resistant to antibiotics, including the 12 families of bacteria most dangerous for human health and for which new antimicrobials are urgently required [3].

It is well known that *A. baumannii* exhibits a wide variety of mechanisms of resistance to antibiotic agents, as differential clones had been isolated in Europe [4]. *A. baumannii* includes several mechanisms of resistance such as lipopolysaccharide expression disorders, permeability alterations due to porins, and the production of active efflux pumps. In particular, resistance to carbapenems is related to numerous beta-lactamases with carbapenemase activity, including type OXA carbapenemases—both constitutive or acquired. Moreover, a transmissible resistance mechanism of colistin, called mobile colistin resistance (MCR), was discovered. Up to ten families with MCR and more than 100 variants of Gram-negative bacteria have been reported worldwide. Even though few have been reported from *Acinetobacter* spp. and *Pseudomonas* spp., it is important to closely monitor the epidemiology of MCR genes in these pathogens [1,4].

*A. baumannii* can survive for long periods on surfaces, including human skin and dry surfaces (up to 33 days) [5], and this ability might facilitate its persistence in intensive care units (ICUs), rightly considered as the epicenters of carbapenem-resistant *A. baumannii* (CR-Ab) infections [6,7]. Some specific factors may increase the potential of cross-transmission: the heavy colonization of the patient, the colonization of the surfaces surrounding the patients, and the total number of patients colonized in the unit at the same time [8]. CR-Ab also has a further important feature, namely its tendency to generate outbreaks, generally transmitted through the hands of healthcare workers, contaminated equipment, and the healthcare environment [7,9,10].

The most frequently reported risk factors for CR-Ab infections are prior colonization, the severity of illness, the need for mechanical ventilation (particularly in case of prolonged duration), immunosuppression, malignancies, chronic pulmonary diseases, respiratory failure on admission, previous antimicrobial therapy, previous sepsis in ICU, previous use of carbapenems and third generation cephalosporins, long ICU stay [11], and a consequent greater degree of exposure to infected or colonized patients in the hospital environment [12,13].

Overall, CR-Ab is accountable for more than 12% of the cases of hospital-acquired bloodstream infections (BSI) in the ICU, with wide geographic variations: it is frequent in Southern Europe, Middle Eastern countries, Asia, and South America, whereas it is rare in Northern European countries and Australia [14]. CR-Ab is a common cause of ICU-acquired pneumonia, particularly late-onset pneumonia [15].

Since the beginning of the COVID-19 pandemic, the impact of superinfections in ICU patients has progressively increased and many studies have shown that the rate of BSIs [16] and Ventilator-Associated Pneumonia (VAP) [17] is raised compared to that observed in non-COVID-19 patients [18,19,20,21]. It has also been reported that the prevalence of Gram-negative multi-drug resistant organisms, especially *A. baumannii*, known to increase mortality, seems to have escalated [22,23].

In Italy, various experiences of multidrug-resistant (MDR) bacterial infection in COVID-19 and its impact on patient outcomes have been published, [24] but few describe specifically *A. baumannii* cases. Several studies showed that MDR infection arose after a median time of 8 [4,5,6,7,8,9,10,11] days and the incidence rate ratio of MDR infection in ICU increased in the COVID-19 period [25,26]. 

Despite the above evidence and the interest in superinfections from multidrug-resistant pathogens, particularly CR-Ab in ICU patients with COVID-19 [27], to date, no multicenter study has been conducted with the aim of better describing the phenomenon.

The present multicenter retrospective study was designed to investigate the characteristics of COVID-19 ICU patients who developed CR-Ab colonization or infection during their ICU stay and evaluate risk factors for ICU mortality in a regional ICU network.

## 2. Materials and Methods

### 2.1. Study Design and Population

This was an observational, multicenter, and retrospective study. Nineteen COVID-19 ICUs in the Piedmont Region, Italy, were invited to participate in an observational, multicenter, retrospective study to describe the incidence of colonization and infection with CR-Ab in SARS-CoV2 pneumonia patients admitted to ICUs between 1 July and 31 December 2021.

The data sources were the hospital administrative records and the Microbiology Laboratory database. Data acquisition and analysis were performed in accordance with the protocols approved by the local Ethics Committee (Ethics Committee: Comitato Etico Interaziendale A.O.U. Città della Salute e della Scienza di Torino—A.O. Ordine Mauriziano—A.S.L. Città di Torino; ethics approval number 0031285). Written informed consent was waived according to Italian regulations due to the retrospective nature of this study. The study was conducted according to the guidelines of the Declaration of Helsinki.

All consecutive adult (≥18 years) patients admitted to the ICU and presenting CR-Ab colonization or infection during the study period were enrolled. All patients were followed up until the hospital discharge to compute: ICU, 28-day and overall mortality, length of ICU and hospital stay.

### 2.2. Context

During the study period, several infection control programs were active in the ICUs involved, with specific leadership and scope. Surveillance cultures (tracheal aspirate, rectal swab, urinary culture) were performed weekly; universal screening for carbapenem-producing *Enterobacterales* (CPE) and *A. baumannii* using rectal swabs was performed upon admission to the ICU and then once a week. In mechanically ventilated patients, the surveillance of respiratory samples (tracheal aspirates or bronchoalveolar lavage) was also performed at least once a week, with some differences between the different centers. Blood cultures or bronchoalveolar lavage cultures were performed on clinical decision.

### 2.3. Definitions

Pneumonia by Severe Acute Respiratory Syndrome Coronavirus 2 (SARS-CoV-2) was defined based on real-time polymerase chain reaction (RT-PCR) on at least one low respiratory tract specimen [28].

The occurrence of colonization and/or infection with *A. baumannii* was assessed from the date of ICU admission to ICU discharge. It was considered only once at the time of the first incidence of a positive sample. Colonization was defined as bacterial isolation without clinical signs or symptoms suggestive of infection. Infection was defined according to the Centers for Disease Control and Prevention (CDC) criteria [29]. Carbapenem resistance was defined according to the EUCAST criteria [30].

All episodes of VAP and/or BSI, as well as the development of septic shock with the requirement of vasoactive drugs [31], were registered according to the European Centre for Disease Prevention and Control (ECDC) current definitions [32].

### 2.4. Microbiology

*A. baumannii* and CPE strains from blood, respiratory, and rectal samples were collected in accordance with active surveillance screening and following local guidelines. Rectal swabs were collected from hospitalized patients and screened for CPE by combining culture-based detection and the identification of carbapenemase type.

We identified CR-Ab according to the European Committee on Antimicrobial Susceptibility Testing (EUCAST) criteria of carbapenem resistance. Cultures were analyzed with the BD BACTECTM FX system (Becton Dickinson) according to EUCAST breakpoint tables. The identification of microorganisms was conducted with mass spectrometry MALDI-TOF (Matrix-Assisted Laser Desorption Ionization Time-of-Flight) and VITEK^®^, whereas susceptibility to antibiotic molecules was tested using VITEK 2 (VITEK^®^ according to EUCAST breakpoint tables). The whole-genome sequencing of CR-Ab isolates collected from blood cultures and respiratory samples was not available in the pandemic context. The clonal relationship of CR-Ab isolates was currently not investigated.

### 2.5. Statistical Analysis

Data were entered and analyzed using SPSS version 27. Statistical significance was defined as less than 0.05. Descriptive analysis was reported as frequencies, percentages, means, and standard deviations. Categorical variables, demographics, and clinical characteristics were compared against mortality using the Chi-squared test. Continuous variables were tested for normality by the Kolmogorov–Smirnov test. Non-normally distributed variables were evaluated using the Mann–Whitney test.

Significant values in the univariate analysis were evaluated with a multivariate model: a logistic regression model for mortality to assess independent predictors. The odds ratio was reported with corresponding 95% confidence intervals.

## 3. Results

Sixteen ICUs joined the data collection. Four of them had no cases of CR-Ab in COVID-19 patients. The first data collection was completed in May 2021. The review of the data by independent investigators was completed in the months of June–September 2021. 

During the study period, 913 COVID-19 patients were admitted to the participating ICUs. Of them, 19% became positive for CR-Ab, either colonization or infection (*n* = 176). The ICU mortality rate in patients with *A. baumannii* was as high as 64.7% (*n* = 112) (Table 1). 

The majority of patients were males (136; 78.6%), with a median age of 65 ± 10.3 years. The average Simplified Acute Physiology Score (SAPS) II and Sequential Organ Failure Assessment (SOFA) scores were 42 ± 13.37 and 8.3 ± 3.7, respectively. Leading comorbidities were cardiovascular diseases (118 patients, 67%), obesity (52 patients, 29.5%), diabetes (39 patient, 22.1%), and chronic pulmonary disease (22 patients, 12.5%). Around 31% of patients were transferred from one hospital to another; 93.2% of them presented acute respiratory distress syndrome upon ICU admission. The mean length of stay in the ICU was 24 ± 18 days. On average, patients developed colonization or infection within 10 ± 8.4 days from ICU admission.

The scores of SAPS II and SOFA were significantly higher in the deceased patients (43.8 ± 13.5, *p* = 0.006 and 9.5 ± 3.6, *p* < 0.001, respectively). Furthermore, the mortality rate was significantly higher in patients with extracorporeal membrane oxygenation (ECMO; 12; 7%, *p* = 0.03), septic shock (61; 35%, *p* < 0.001), and in elders (66 ± 10, *p* < 0.001) (Table 1). 

Among the 176 patients enrolled in the study, 129 (73%) had invasive infection with CR-Ab, distributed as follows: 105 (60.7%) VAP and 46 (26.6%) BSI. In 22 cases (6.5%), VAP was associated with concomitant BSI. Colonization was reported in 165 patients (93.7%). Of note, 118 patients previously colonized by CR-Ab developed invasive infections, while 11 patients developed infection without any known previous colonization. Mortality was significantly higher in patients with VAP (*p* = 0.009). Colonized patients who did not develop invasive infections had a higher survival rate (*p* < 0.001; Table 1). Being colonized by CR-Ab was associated with a higher risk of developing invasive infections (*p* < 0.001). 

Co-infections with carbapenem-resistant Klebsiella pneumoniae and enteric pathogens were seen in 29 (17%) and 55 (32%) patients, respectively. 

Most of the CR-Ab isolates (159, 90.3%) were sensitive to colistin. Colistin was used to treat the majority (100, 56.8%) of the patients. Most commonly, it was administered with meropenem, ampicillin-sulbactam, and rifampicin (21%, 18.1%, and 17%), respectively. However, no difference in mortality rate was observed between different therapies. 

In the multivariate analysis (Table 2), risk factors that were significantly associated with mortality were age (OR = 1.070; 95% CI (1.028–1.115) *p* = 0.001) and CR-Ab colonization (OR = 5.463 ic 96% 1.572–18.988 *p* = 0.008). 

## 4. Discussion

Bacterial and fungal superinfections represent a severity factor with a high impact on the morbidity and mortality of critically ill patients with COVID-19 [33,34]. This aspect is even more essential in countries burdened by a high rate of multidrug-resistant bacteria, such as Italy [35], where an increasing number of CR-Ab infections have been seen in the last years. 

In the present multicentric study, conducted on 16 ICUs in the Piedmont region during the COVID-19 pandemic, it was found that 19% of ICU COVID-19 patients became positive for CR-Ab, either colonization or infection, during an ICU stay. Although the whole-genome sequencing of CR-Ab isolates was not available in the pandemic context and the clonal relationship of CR-Ab isolates was currently not investigated, this elevated percentage and some epidemiological factors deserve very high attention. Furthermore, the mortality rate in patients with CR-Ab was as high as 64.7%, significantly higher than the overall mortality in critically ill COVID-19 patients [36].

To the best of our knowledge, this is the first multicenter regional study reporting the impact of CR-Ab colonization and severe infection in ICUs during the COVID-19 pandemic. Interestingly, our analysis refers to the so-called Italian “second-wave” of the pandemic, when the global emergency scenario of the first months of the pandemic had extensively changed. A recent multicenter, cross-sectional study compared the rates of colonization and infection with carbapenemase-producing *Enterobacterales* (CPE) and/or CR-Ab in two study periods, pre and during the COVID-19 pandemic. No significant change in either incidence rate ratios and weekly trends in CPE colonization and infection was observed, while the incidence rate ratios of colonization and infection with CR-Ab increased by 7.5- and 5.5-fold, respectively, during the COVID-19 period. A clonal lineage was demonstrated and appointed for the occurrence of horizontal transmission [26].

Other authors previously highlighted that, during the first wave of the COVID-19 pandemic, several factors could have favored the emergence and spread of antimicrobial-resistant bacteria in hospitals [25], such as the overload of hospitalized patients, especially in intensive care, favoring patient-to-patient transmission [37]; the initial overuse of antibiotics for suspected bacterial co/super-infections [38]; the possible delay in providing microbiological culture and sensitivities results due to the COVID-19 overload [39]. During the first months of the pandemic, in several countries, including Italy, a lack of appropriate protective personal equipment and health personnel hired on an emergency basis to respond to the COVID-19 pandemic, sometimes impeding adequate training in infection prevention and control, were common. However, that may not be completely true in the period of our study, when the first pandemic phase with its need for reorganization was already over.

Other factors may have contributed to the described spread of CR-Ab infections.

First of all, the need for the referral of critically ill patients (e.g., requiring ECMO [40]) and the high number of patients transferred from one hospital to another may have facilitated the dissemination of cases at the regional level. Even the structural characteristics of ICUs (new, re-opened, or already functioning before the COVID-19 pandemic) may also have played a role, in terms of spaces dedicated to patients and workstations, devices, and hospital pathways between departments (e.g., emergency department, radiology). In fact, CR-Ab cross-transmission between equipment (ventilators, infusion pumps, hemodialysis machines, ultrasound devices) and COVID-19 patients may also partly explain the onset of this outbreak.

Focusing on the identification and characterization of Enterococcus faecium, Staphylococcus aureus, Klebsiella pneumoniae, *Acinetobacter baumannii*, Pseudomonas aeruginosa, and Enterobacter spp. (ESKAPE) bacteria and their possible clonal spread in medical devices, patients, and medical personnel in the ICU, a recent work [41] has shown that 91% of the analyzed sites were colonized by bacteria (pathogenic and commensal), where *S. aureus* and *A. baumannii* MDR showed a high incidence, and *A. baumannii* MDR showed a clonal distribution in surfaces, patients, and health personnel.

It is in fact known that even when there is the scrupulous protection of medical personnel to avoid the transmission of SARS-CoV-2 from patients to health personnel, the transmission of other pathogens such as ESKAPE bacteria is not automatically avoided. In a previous study in ICUs, it was shown that the bacterial recontamination of contact surfaces occurred after 4 h after standard cleaning with detergents with chlorine-releasing agents, isopropyl alcohol, and sodium hypochlorite [42]. Moreover, COVID-19 critically ill patients often require prolonged hospitalizations, and it is known that staying in an intensive care setting for a long time—as well as immunosuppression, the need for prolonged previous antibiotic therapies, and the invasiveness of care—are known risk factors for infections with multidrug-resistant pathogens. 

In our analysis, the median ICU length of stay was high (24.27 ± 17.9 days), with a time lag before the development of colonization or the onset of invasive infection of 17.31 ± 13.3 days of hospital stay and 10.69 ± 8.4 days of ICU stay, respectively. 

Some other factors must be taken into account in the analyzed population. Certainly, patient severity had an impact on mortality, with statistical significance for the need for ECMO support, higher SAPS and SOFA scores, and the presence of septic shock as infection presentation. Similar to other settings, the use of steroids might be related to a higher risk of developing MDR infection [43]. Concerning the impact of VAP in CR-Ab infected patients, the diagnosis of VAP may have been made difficult by the presence of the radiological and clinical signs of COVID-19 pneumonia, which made it even more difficult to apply the classical criteria and the consequent definition of VAP.

The presence of colonization preceding the infection represented, in our series, a risk factor with respect to mortality. It is well known from the literature that colonization does not require any “pre-emptive” therapy if the patient has no clinical signs of infection, but these data confirmed the finding that colonization remains one of the main risk factors for invasive infections and represent a “wake up call” regarding the frailty of our patients. Therefore, implementing an early pre-emptive therapy in cases of known colonization, at the time of clinical worsening, is one of the main steps to improve survival in this setting.

As previously reported in the literature, the role of combination therapy is widely debated in the absence of definitive evidence [44,45]. The data are insufficient for a more completed analysis, but the unmet need for new and effective therapies is of paramount importance considering the mortality of these critically ill patients.

The presence of multi-bacterial co-infections is a further interesting fact, able to describe not only the fragility of the patients but also the delicate hospital ecology and to reinforce the need for effective and strict control measures. In particular, the combination of various Gram-negative pathogens describes the context of our ICUs and may be the consequence of the high use of empiric broad-spectrum antibiotic therapies used in COVID patients not only at home but also in the early stages of hospitalization.

Our study has several limitations. First, the retrospective nature of the study and therapeutic management on the risk of *A. baumannii* infection. Secondly, the lack of data on the total number of COVID-19 ICU patients did not allow a comparison of risk factors and outcomes. Third, as the clonal relationship was not investigated, it is impossible to define the common origin of the burden of infections or a relationship, at least in the high number of referral patients. Moreover, it was not possible to obtain a cumulative antibiogram for antibiotic classes to show the overall sensibility of different strains. Finally, the local epidemiology and the need to re-organize the capacity, spaces, and staff of our ICUs during the pandemic could limit the generalizability of our results.

## 5. Conclusions

The need to not neglect antimicrobial stewardship principles during the COVID-19 pandemic has already been recently underlined [46], as well as the importance of enhancing infection control activities directed against antimicrobial resistance. In continuity with this message, our study remarks on the need to pursue antimicrobial stewardship principles during the COVID-19 pandemic, and infection control activities targeted against the spread of antimicrobial resistance inside and between hospitals.

During a pandemic, not only in the first phases, but especially later in the time course, infection control activities should be revised and eventually re-modulated according to the new organizational structures. Constant infection-control measures are necessary to stop the spread of *A. baumannii* in the hospital environment, prevent outbreaks, and lower mortality rates, especially at this time of the SARS-CoV-2 pandemic. Stricter barrier measures need to be implemented, increasing the effectiveness of screening and surveillance for *A. baumannii*, especially when resistant to carbapenems. The active surveillance culture and efficient performance of a multidisciplinary team will be highly important in detecting and controlling the CR-Ab outbreak in COVID-19 ICUs.

## Figures and Tables

**Table 1 jcm-11-05208-t001:** Demographic and general characteristics of COVID-19 ICU patients with CR-Ab.

Variable (*n*; %)	Total 176 (100%)	Survived 61 (35.3%)	Dead 112 (64.7%)	*p*-Value
**Demographics**				
Males	136 (78.6)	48 (67.6%)	88 (78.6%)	0.986
Age	65.35 ± 10.3	62.84 ± 10.7	66.44 ± 10	**<0.001**
BMI	30.8 ± 7.3	31.33 ± 7.4	30.83 ± 7.36	0.858
Ex-smoker	8 (4.5)	4 (6.5)	4 (3.5)	0.372
Smoker	8 (4.5)	3 (4.9)	5 (4.5)	0.892
Obese	52 (29.5)	20 (32.8)	32 (28.6)	0.563
**Comorbidities**				
Cardiovascular disease	118 (67)	38 (62.3)	80 (71.4)	0.218
Diabetes	39 (22.1)	12 (19.7)	27 (24.1)	0.505
Hematologic disease	2 (1.1)	1 (1.6)	1 (0.9)	0.661
Chronic pulmonary disease	22 (12.5)	6 (9.8)	16 (14.3)	0.401
Renal failure	15 (8.5)	5 (8.1)	10 (8.9)	0.870
Active neoplasm	7 (4)	2 (3.3)	5 (4.5)	0.705
Autoimmune disease	18 (10.2)	6 (9.8)	12 (10.7)	0.857
Immunodepression	4 (2.3)	2 (3.3)	2 (1.8)	0.532
**Clinical characteristics**				
ICU length of stay	24.27 ± 17.9	25.7 ± 20.58	24.1 ± 18.22	0.930
Days to infection/colonization from hospital admission	17.31 ± 13.3	17.2 ± 13.44	17.31 ± 12.34	0.718
Days to infection/colonization from ICU admission	10.69 ± 8.4	10.63 ± 8.38	10.69 ± 8.42	0.585
Referral	54 (30.7)	17 (27.9)	37 (33)	0.483
ECMO	13 (7.4)	1 (1.6)	12 (10.7)	**0.031**
SAPS II	42.28 ± 13.37	41.6 ± 13	43.88 ± 13.5	**0.006**
SOFA	8.3 ± 3.7	6 ± 2.6	9.5 ± 3.6	**<0.001**
ARDS on admission	165 (93.2)	59 (96.7)	106 (94.6)	0.534
Septic shock	67 (38.1)	6 (9.8)	61 (54.5)	**<0.001**
Colistin sensitive	159 (90.3)	53 (86.9)	106 (94.6)	0.074
Colistin resistant	14 (7.9)	8 (13.1)	6 (5.3)	
Carbapenem-resistant	122 (69.3)	46 (75.4)	76 (67.8)	0.479
**Invasive infections**				
CR-Ab VAP	105 (59.6)	29 (47.5)	76 (67.8)	**0.009**
CR-Ab BSI	46 (41.1)	14 (22.9)	32 (28.6)	0.424
CR-Ab + **co-infection**				
*K. pneumoniae*—KPC	29 (16.5)	11 (18)	18 (16.1)	0.726
MRSA	8 (4.5)	3 (4.9)	5 (4.5)	0.892
VRE	7 (4)	3 (4.9)	4 (3.6)	0.668
Enteric pathogens	55 (31.2)	18 (29.5)	37 (33)	0.634
**Colonization**				
CR-Ab	165 (93.7)	58 (95)	104 (92.8)	0.567
Cp-*K.pneumoniae*	14 (7.9)	4 (6.5)	10 (8.9)	0.585
VRE	1 (0.6)	1 (1.6)	0	0.174
*E.coli*	2 (1.1)	2 (3.2)	0	0.054
*Candida* spp	8 (4.5)	3 (4.9)	5 (4.5)	0.892
MRSA	5 (2.8)	1 (1.6)	4 (3.6)	0.469
Other	74 (42)	28 (45.9)	46 (41.1)	0.587
**Combination treatment with colistin**				
Total colistin treatment	100 (56.8)	33 (54)	67 (59.8)	0.466
Colistin monotherapy	10 (5.7)	1 (1.6)	9 (8)	0.085
Meropenem	37 (21)	16 (26.2)	21 (18.7)	0.252
Ampicillin sulbactam	32 (18.1)	9 (14.7)	23 (20.5)	0.349
Rifampicin	30 (17)	9 (14.7)	21 (18.7)	0.507
Tigecycline	15 (8.5)	2 (3.2)	13 (11.6)	0.063
Vancomycin	7 (4)	3 (4.9)	4 (3.6)	0.668
Ceftazidime-avibactam	7 (4)	1 (1.6)	6 (5.3)	0.236
**Only colonized/infected vs. mortality**				
CR-Ab colonized (without infection)	47 (26.7)	25 (40.9)	22 (19.6)	**<0.001**
CR-Ab infected (without colonization)	11 (6.2)	3 (4.9)	8 (7.1)	0.567

List of abbreviations: intensive care unit, ICU; carbapenem-resistant *Acinetobacter baumannii*, CR-Ab; number, *n*; Body Mass Index, BMI; extracorporeal membrane oxygenation, ECMO; Simplified Acute Physiology Score, SAPS; Sequential Organ Failure Assessment, SOFA; Acute Respiratory Distress Syndrome, ARDS; Ventilator-Associated Pneumonia, VAP; Bloodstream infection, BSI; *K.pneumoniae* producing KPC; methicillin-resistant *Staphylococcus aureus*, MRSA; vancomycin-resistant *Enterococcus*, VRE. bold was used for *p* < 0.05.

**Table 2 jcm-11-05208-t002:** Multivariate analysis for mortality.

Variables	*p* Value	OR	95% C.I.for EXP(B)	
			Lower	Upper
**Age**	0.001	**1.070**	1.028	1.115
**SAPS II**	0.145	1.022	0.992	1.053
**VAP**	0.451	1.568	0.487	5.049
**CR-Ab** **colonization**	0.008	**5.463**	1.572	18.988

List of abbreviations: Simplified Acute Physiology Score, SAPS; Ventilator-Associated Pneumonia, VAP; carbapenem-resistant *Acinetobacter baumannii*, CR-Ab. bold was used for *p* < 0.05.

## Data Availability

The datasets used and analyzed during the current network meta-analysis are available from the corresponding author upon reasonable request.

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
