# Peer review of "The Burden of Carbapenem-Resistant Acinetobacter baumannii in ICU COVID-19 Patients: A Regional Experience"

_jcm, 2022, doi:10.3390/jcm11175208_

Round 1
Reviewer 1 Report
The manuscript titled “The Burden of Carbapenem Resistant Acinetobacter baumannii in ICU COVID-19 patients: A Regional Experience” describes the observational multicenter retrospective study of CRAB in ICU COVID-19 patients. This manuscript is very interesting because the study shed the light on the relationship between CRAB infection and COVID-19 patients. The authors provided sufficient data to support their conclusions, but a few clarifications should be supplied. Therefore, I recommend this manuscript is accepted by minor revisions.
1. Method: The authors should clarify how was CRAB identified in this study?
2. Please check A. baumannii. It should be italicized
Author Response
The manuscript titled “The Burden of Carbapenem Resistant Acinetobacter baumannii in ICU COVID-19 patients: A Regional Experience” describes the observational multicenter retrospective study of CRAB in ICU COVID-19 patients. This manuscript is very interesting because the study shed the light on the relationship between CRAB infection and COVID-19 patients. The authors provided sufficient data to support their conclusions, but a few clarifications should be supplied. Therefore, I recommend this manuscript is accepted by minor revisions.
Thank you very much for your review, for appreciating our work and for your help in improving it.
- Method: The authors should clarify how was CRAB identified in this study?
Thank you very much for rising this important point. As suggested, we add this sentence to the methods section:
“We identify CR-AB according to European Committee on Antimicrobial Susceptibility Testing (EUCAST) criteria of Carbapenem resistance. Cultures were analyzed with BD BACTECTM FX system (Becton Dickinson) according to EUCAST breakpoint tables. Identification of microorganism was conducted with mass spectrometry MALDI- TOF (Matrix Assisted Laser Desorption Ionization Time-of-Flight) and VITEK® whereas susceptibility to antibiotic molecules was tested using VITEK 2 (VITEK® according to EUCAST breakpoint tables).”
- Please check A. baumannii. It should be italicized
Thanks for this suggestion, we corrected the draft accordingly.
Reviewer 2 Report
Given the intensive care profile of the Journal, the study covers the most important clinical findings related to CR Acinetobacter. Nevertheless, minimal microbiological details should be included, for better understanding of this topic by clinician readers.
Introduction section is too long. Lines 109-121 - can be passed to discussions. Basic information regarding the mechanisms of carbapenem resistance should be presented (efflux pumps, carpanemenase enzyme production), which will give a better picture on the actual threat of antibiotic resistance and on the chance of horizontal resistance gene transmission. Also, as Table 1 presents colistin-related findings, data on Acinetobacter colistin resistance should be presented in introduction (including the chance of mcr-1 plasmid gene, which is easily transmittable between bacteria).
Discussions - please address the topic: colonization vs infection in Acinetobacter and the associated therapeutic protocols .
Line 177 - clonal relationship can be partially assessed by antibiotic susceptibility pattern.
Cumulative antibiogram for antibiotic classes can be easily calculated and should be presented, which will add value to the study. This will show the overall susceptibility/resistance to other classes of antibiotics.
Author Response
Given the intensive care profile of the Journal, the study covers the most important clinical findings related to CR Acinetobacter. Nevertheless, minimal microbiological details should be included, for better understanding of this topic by clinician readers.
Thank you very much for your review, for appreciating our work and for your help in improving it.
Introduction section is too long. Lines 109-121 - can be passed to discussions.
Thank you for this advice. We change the draft accordingly, in particular:
In the introduction, we syntetized as follow: “Several studies showed that MDR infection arised after a median time of 8 [4–11] days and the incidence rate ratio of MDR infection in ICU increased in COVID-19 period [25-26]. “
Then, we postponed in the discussion: “A recent multicenter cross-sectional study compared the rates of colonization and infection with carbapenemase producing Enterobacterales (CPE) and/or CR-Ab in two study periods, pre and during COVID-19 pandemic. No significant change in either incidence rate ratios and weekly trend in CPE colonization and infection was observed, while the incidence rate ratios of colonization and infection with CR-Ab increased of 7.5 and 5.5-fold, respectively, during COVID-19 period. A clonal lineage was demonstrated and appointed for the occurrence of horizontal transmission [26].”
We removed the sentence: “In a small, retrospective analysis on ICU patients, fifty percent of patients developed carbapenem-resistant Klebsiella pneumoniae and A. baumannii, MDR infection after a median time of 8 [4–11] days. This has been reported to be associated with a longer length of ICU stay (p = 0.002), a higher use of steroid therapy (p = 0.011) and lower ICU mortality (odds ratio: 0.439,95% confidence interval: 0.251–0.763; p < 0.001) [25].”
Basic information regarding the mechanisms of carbapenem resistance should be presented (efflux pumps, carpanemenase enzyme production), which will give a better picture on the actual threat of antibiotic resistance and on the chance of horizontal resistance gene transmission.
Thank you for this important suggestion. We combined this point with the following (see below).
Also, as Table 1 presents colistin-related findings, data on Acinetobacter colistin resistance should be presented in introduction (including the chance of mcr-1 plasmid gene, which is easily transmittable between bacteria).
Thanks for your suggestion, we have added a sentence in the introduction that better explain the different resistance mechanisms of A.baumannii:
“A. baumannii includes several mechanisms of resistance as lipopolysaccharide expression disorders, permeability alterations due to porins and production of active efflux pumps. In particular, resistance to carbapenems in A. baumannii is related to numerous beta-lactamases with carbapenemase activity, including type OXA carbapenemases - both constitutive, or acquired. Moreover, a transmissible resistance mechanism of colistin, called mobile colistin resistance (MCR), was discovered. Up to ten families with MCR and more than 100 variants of Gram-negative bacteria have been reported worldwide. Even though few have been reported from Acinetobacter spp. and Pseudomonas spp., it is important to closely monitor the epidemiology of MCR genes in these pathogens.”
Discussions - please address the topic: colonization vs infection in Acinetobacter and the associated therapeutic protocols.
Thank you for raising this important point. As widely indicated in the literature, only the cases of infection were treated, while in the case of colonization contact isolation protocols were put in place, but no antimicrobial therapies. In cases in which patients already identified as colonized have developed signs and symptoms of invasive infection, the possible use of specific therapy for A.baumannii (pre-emptive therapy) has been evaluated on the basis of the clinical judgment of involved clinicians, often assisted by infectious disease consultants.
We added a sentence in the discussion to better specify this concept: “It is well known from literature that colonization does not require any “pre-emptive” therapy if the patient has no clinical signs of infection, but this data confirmed the finding that …”
Line 177 - clonal relationship can be partially assessed by antibiotic susceptibility pattern.
Cumulative antibiogram for antibiotic classes can be easily calculated and should be presented, which will add value to the study. This will show the overall susceptibility/resistance to other classes of antibiotics.
Thank you for this interesting proposal. Certainly, the point of view of the rev. 2 opens to a possible deepening of the resistance data, at least of the phenotypic type, on the basis of the antibiograms.
However, as this was a retrospective multicenter study, unfortunately the data requested from the various centers did not include the complete antibiogram, but only the definition of resistance to specific patterns of interest (such as R to Cp and colistin, as defined in the tables).
Therefore, it is now not possible for us, a posteriori and within a reasonable time, to obtain the requested data. Surely this aspect will be taken into account by the authors in case of further investigations and studies on these aspects. We added this concept as a limitation of our study.
“Moreover, it was not possible to get a cumulative antibiogram for antibiotic classes to show the overall sensibility of different strains.”